# Source-Free Domain Adaptation Framework for Rotary Machine Fault Diagnosis

**DOI:** 10.3390/s25144383

**Published:** 2025-07-13

**Authors:** Hoejun Jeong, Seungha Kim, Donghyun Seo, Jangwoo Kwon

**Affiliations:** 1Department of Electrical and Computer Engineering, Inha University, Incheon 22212, Republic of Korea; lilmae@inha.edu (H.J.); tmdgk4871@inha.edu (S.K.); korsdh@inha.edu (D.S.); 2Department of Computer Engineering, Inha University, Incheon 22212, Republic of Korea

**Keywords:** fault diagnosis, domain adaptation, variational autoencoder (VAE), self-supervised learning, rotating machinery, test-time training (TTT)

## Abstract

Intelligent fault diagnosis for rotary machinery often suffers performance degradation under domain shifts between training and deployment environments. To address this, we propose a robust fault diagnosis framework incorporating three key components: (1) an order-frequency-based preprocessing method to normalize rotational variations, (2) a U-Net variational autoencoder (U-NetVAE) to enhance adaptation through reconstruction learning, and (3) a test-time training (TTT) strategy enabling unsupervised target domain adaptation without access to source data. Since existing works rarely evaluate under true domain shift conditions, we first construct a unified cross-domain benchmark by integrating four public datasets with consistent class and sensor settings. The experimental results show that our method outperforms conventional machine learning and deep learning models in both F1-score and recall across domains. Notably, our approach maintains an F1-score of 0.47 and recall of 0.51 in the target domain, outperforming others under identical conditions. Ablation studies further confirm the contribution of each component to adaptation performance. This study highlights the effectiveness of combining mechanical priors, self-supervised learning, and lightweight adaptation strategies for robust fault diagnosis in the practical domain.

## 1. Introduction

Rotating machinery plays a vital role across various industrial sectors such as manufacturing, power generation, and transportation, serving as key equipment for transmitting power and converting energy. Due to their continuous rotational motion, components such as shafts, bearings, and gears are prone to wear, imbalance, and mechanical looseness. If such anomalies are not detected early, they may lead to equipment failure, production downtime, or even safety accidents. Therefore, fault diagnosis technologies capable of the real-time monitoring and early detection of abnormal behavior are essential to ensure the reliability and productivity of industrial sites.

Vibration signals directly reflect changes in the physical condition of internal components and are capable of capturing fault indications through variations in energy distribution across specific frequency bands. For this reason, vibration signals have been established as a standard input for rotary machinery diagnostics, and numerous studies have reported high classification accuracy for various fault types using these signals [1,2].

Traditional vibration-based fault diagnosis typically involves extracting handcrafted features using signal processing techniques such as FFT (Fast Fourier Transform), EMD (Empirical Mode Decomposition), or Order Tracking, followed by classification using conventional machine learning algorithms such as SVM or decision trees [3,4]. While such approaches can achieve high performance under specific conditions, they heavily depend on expertly designed features and are vulnerable to performance degradation when sensor positions or environmental conditions change. To overcome these limitations, recent studies have introduced deep learning-based models such as CNNs and RNNs, which automate feature extraction and improve classification performance through label-based training [5,6,7].

However, most existing deep learning approaches rely on single-input classification architectures, which often result in overfitting to specific training domains without learning sufficiently generalized representations. To address this structural limitation, we propose a self-supervised framework that reconstructs the input vibration signal to learn a generalized representation and then classifies anomalies by comparing the latent representations of the input and reference signals. This framework goes beyond conventional classification by interpreting the structural differences between input and reference data in latent space, enabling more robust fault diagnosis under domain shifts.

Furthermore, to improve reconstruction accuracy, we designed an encoder–decoder architecture based on U-Net [8] instead of a conventional CNN-based autoencoder. U-Net’s skip connections help preserve multi-resolution features, allowing for the finer reconstruction of frequency details and more precise latent representation learning. This architecture supports self-supervised training even in the absence of labels by minimizing reconstruction loss, thereby providing adaptability to target domains without additional annotation. Since, in many industrial settings, access to source domain data is restricted and labeled data from the target domain is often unavailable, we adopt a Source-Free Domain Adaptation (SFDA) strategy and verify its effectiveness experimentally.

The main contributions of this paper are as follows: (1) We propose an order-frequency-based preprocessing method that normalizes rotational variations across machines, enabling unified representation learning in cross-domain vibration diagnostics. (2) We develop a self-supervised anomaly classification framework based on a U-Net-structured variational autoencoder, which reconstructs input signals and compares latent representations with reference signals for robust fault detection. (3) We introduce a test-time training strategy tailored for Source-Free Domain Adaptation, allowing our model to adapt to unlabeled target domains without access to source data. A unified benchmark of four public datasets is constructed to validate performance under real-world domain shift conditions.

## 2. Related Work

### 2.1. Rotary Machine Fault Diagnosis

Faults in rotary machinery are directly linked to equipment downtime, product quality degradation, and financial losses. As such, fault diagnosis technologies that enable early detection and response have played a critical role in industrial applications. In particular, vibration signals have been widely utilized as key indicators for diagnosing the condition of rotating equipment, as they quantitatively capture subtle dynamic variations. Since many fault symptoms appear as patterns in the frequency or time–frequency domain, vibration-based diagnostics have proven highly effective.

Traditional fault diagnosis approaches initially relied on physics-based models or statistical process control techniques to assess the health of rotating machinery. For instance, Farrar et al. [9] proposed a method based on statistical pattern recognition, assuming that minor changes in structural dynamic responses reflect the presence and severity of damage. Their method involved training an autoregressive (AR) model on undamaged signals, then monitoring the mean and variance of the residuals using control charts for damage detection. While such approaches provide interpretable and quantitative diagnostic criteria, they are often sensitive to nonlinearities and external factors such as load variations and temperature changes, limiting their generalizability in real-world operating environments.

With the introduction of machine learning techniques, diagnostic pipelines began to incorporate signal processing methods such as FFT and EMD for feature extraction, followed by classifiers like SVM or ANFIS [10]. For example, Lei et al. [11] used EMD to extract intrinsic mode functions, selected meaningful features using an enhanced distance metric, and fed them into multiple ANFIS classifiers for fault prediction. They further improved classification performance under multiple fault conditions by combining the outputs using a genetic algorithm. Although these methods improved automation and accuracy over statistical approaches, they still required manual feature selection and were prone to performance degradation when applied to different operating environments.

More recently, deep learning-based methods have gained prominence due to their ability to extract high dimensional representations directly from raw vibration signals without explicit preprocessing or feature engineering. Zhang et al. [12] proposed the WDCNN model, which uses wide kernels in the initial layers to suppress high-frequency noise and smaller kernels in subsequent layers to capture fault-specific patterns. This end-to-end approach achieved high classification accuracy and demonstrated robust performance even on real industrial datasets.

Vibration signals are inherently non-stationary and time-varying, so raw time-series inputs cannot capture consistent fault features across speed and load changes. Shao et al. [13] showed that STFT-based time–frequency spectra provide stable fault patterns under varying conditions. Aiordăchioaie [14] further demonstrated that an FFT-based power spectrum achieves higher classification accuracy rather than time-domain methods. However, even frequency-domain inputs shift with RPM variation. Wang et al. [15] used order analysis to normalize rotational speed effects and produce order-domain features that outperform both raw and standard frequency inputs under variable speeds. Yu et al. [16] then resolved the spectral blur of conventional FFT-order spectra by using mean envelope amplitudes to yield high-resolution, noise-robust order spectra. In line with these findings, we apply an FFT followed by an order transform to supply our model with speed-invariant, high-resolution frequency representations.

### 2.2. Self-Supervised Feature Extraction via VAE

Vibration signals measured from rotary machinery vary in distribution depending on external factors such as rotational speed, load conditions, sensor position, and ambient temperature. Even for the same type of fault, the signal may appear differently under different conditions. To perform reliable diagnostics under such complex variations, it is essential to learn a universal representation that effectively captures the inherent structural characteristics of the input data.

In recent studies, self-supervised learning approaches that reconstruct the input signal itself have been actively explored to overcome the limitations of supervised classification models [17,18,19]. In particular, the VAE has been regarded as a highly suitable architecture for this purpose.

The VAE [20] embeds the input into a probabilistic latent space and reconstructs it based on the sampled latent representation. It is known to learn more generalized representations than a standard autoencoder. Bengio et al. [21] experimentally demonstrated that AE/VAE-based reconstruction learning can provide meaningful representations for downstream tasks, and emphasized the mathematical superiority of VAE in learning the overall data distribution rather than merely compressing features. Since the VAE can learn structured representations of input data without supervision, it is advantageous for building models that are robust to various domain conditions.

Figure 1 visually illustrates the difference between a standard autoencoder (AE) and a variational autoencoder (VAE). While an AE compresses and reconstructs the input deterministically, a VAE maps the input into a latent space modeled as a Gaussian distribution and reconstructs it through sampling. This probabilistic structure improves both the diversity and generalizability of the learned representations.

VAEs have been successfully applied not only in image domains but also in sensor-based time-series data. In particular, they have been utilized in temporal representation learning as an effective method to capture essential patterns within input sequences. For example, when applied to one-dimensional time-series data such as vibration signals, VAE-based reconstruction enables the model to learn high-dimensional representations that include regions containing critical fault information. These representations can be effectively used for downstream tasks such as classification and anomaly detection [22,23].

Recent studies have further extended this reconstruction-based representation learning to multi task structures, where reconstruction and classification are trained simultaneously [24,25]. By jointly optimizing both tasks, the model is encouraged to capture the structural characteristics of the input beyond relying solely on label information, thereby improving generalization under domain shift. This learning paradigm is particularly effective in test-time training (TTT) and Source-Free Domain Adaptation (SFDA) scenarios, where adaptation must be performed without labeled data.

In this study, we adopt such a self-supervised representation learning framework. Our model reconstructs vibration frequency data using a VAE, while simultaneously learning generalized latent vectors. Anomalies are then classified by comparing the latent representation of the input with that of a reference (normal) state. The reconstruction loss is optimized without any labels, and the classification based on latent differences enables robust fault diagnosis even under domain shifts. In this study, we adopt a standard Gaussian prior for the VAE latent space. While alternative priors such as mixture models or β-VAE can offer more expressive latent distributions, they typically require access to additional information such as class labels or structural attributes. However, our framework is designed for source-free and unsupervised domain adaptation, where neither source data nor target labels are available during adaptation. In this setting, the use of a standard normal prior is the most practical and stable choice, allowing the model to learn transferable latent features without relying on any domain-specific assumptions.

Our model reconstructs vibration frequency data using a VAE with a standard Gaussian prior in the latent space, which enables it to learn generalized representations without labels. This choice is particularly suitable for our source-free and unsupervised setting, where additional information required by more expressive priors (e.g., β-VAE [26]) is unavailable. Fault conditions are then identified by comparing the latent representation of the input with that of a reference (normal) state, allowing robust diagnosis across domains with different operating conditions.

### 2.3. Source-Free Domain Adaptation

Domain adaptation is a core technique for ensuring that a model performs reliably even when deployed in environments with data distributions different from those used during training. It is considered essential for real-world industrial applications. However, most existing domain adaptation methods assume simultaneous access to labeled source domain data and unlabeled target domain data. This assumption often fails in practice, where accessing source data is restricted due to security, privacy regulations, intellectual property, or legal constraints.

Source-Free Domain Adaptation (SFDA) has recently gained attention as a method to address this limitation. SFDA enables model adaptation using only a pre-trained source model and unlabeled target domain data, without requiring access to the original source data. This paradigm provides a practical solution that satisfies both data privacy and model portability requirements, making it particularly suitable for industrial deployment scenarios.

Figure 2 illustrates the general structure of Source-Free Domain Adaptation (SFDA). A pre-trained model consisting of a feature extractor and classifier is kept frozen or partially adaptive, and adaptation is performed using only unlabeled data from the target domain. This process aims to either reduce the distributional gap or reorganize the representation space to better align with the target data. Common approaches include entropy minimization with dispersion regularization to sharpen decision boundaries, or iterative refinement using soft pseudo-labels generated via k-means clustering [27]. Other methods introduce self-supervised auxiliary tasks, such as predicting image rotation or position, to guide adaptation to the new distribution [28], or apply contrastive learning to preserve feature distances between augmented samples during domain shifts [29].

While these techniques originated mainly in the image domain, recent studies have begun to leverage reconstruction-based self-supervised learning to align cross-domain representations at test time, even without labels [30]. In addition, student–teacher frameworks have been introduced to improve prediction stability and enable gradual adaptation without relying heavily on pseudo-labels [31]. Collectively, these efforts reflect an evolution of SFDA toward preserving classification performance while removing the dependency on labeled or source data.

However, applying SFDA to time-series data where sequential continuity and temporal dependency are critical presents additional challenges. Unlike images, the temporal structure often varies significantly across domains, making simple alignment strategies insufficient. To address this, some recent work has proposed masking parts of the time series and training the model to reconstruct them, thereby learning the temporal structure of the target domain in a self-supervised manner [32]. Other approaches train autoregressive interpolation models on the source domain and enforce temporal consistency when applied to the target domain [33]. In time-series image-based domain adaptation, methods have also been proposed to balance class separability and information preservation while aligning cross-domain features [34].

In addition to approaches that rely solely on deep learning, there has been a growing interest in hybrid frameworks that incorporate physical signal preprocessing and domain knowledge into the domain adaptation process. For example, ref. [35] explores the fusion of real data acquired from an actual system with synthetic data generated from a physical model, generating synthetic data for every possible condition. Ref. [36] explores using simulation-based training data to overcome the challenges of limited real-world fault data for bearing fault detection, enabling machine learning models to be trained for fault detection even in the absence of sufficient labeled fault data. Furthermore, ref. [37] proposes a zero-fault-shot learning framework that combines physical preprocessing with model-based approaches to achieve robust domain adaptation for bearing spall type classification. This method demonstrates that integrating domain-specific physical priors with machine learning models can significantly improve adaptation performance.

In this study, we extend existing time-series SFDA approaches by constructing a fault diagnosis framework that combines reconstruction-based self-supervised learning with latent space comparison. Specifically, we reconstruct the representation space and classify anomalies based on the latent difference between the input and a reference (normal) signal, enabling effective adaptation to the target domain without labels.

## 3. Method

### 3.1. Mechanically Informed Order Spectrum Preprocessing

Vibration signals measured from rotary machinery exhibit distinctive frequency-domain patterns when mechanical faults or structural anomalies occur. In particular, depending on the type of fault, energy tends to concentrate at integer multiples of the fundamental rotational frequency. Since these patterns are dependent on rotational speed, a change in speed results in a shift in the fault-related frequencies, making consistent fault identification across varying operating conditions difficult. This variability becomes a key reason why deep learning models may learn the same fault under different distributions, ultimately hindering generalization.

To address this issue, the mechanical engineering community has traditionally employed a technique known as order analysis. Order analysis normalizes the frequency axis based on the machine’s rotational speed (RPM), aligning fault features at consistent order indices regardless of operating speed. This approach has long been validated in industrial settings and is considered a standard frequency-domain diagnostic method. Compared to conventional FFT-based analysis, order analysis provides superior consistency in diagnosing the same fault under variable speed conditions.

Despite its effectiveness, most recent deep learning-based diagnostic studies have bypassed such physically grounded preprocessing steps, instead relying solely on raw time-domain signals or standard transformations such as FFT [38], Short-Time Fourier Transform (STFT) [39], or wavelet transform (WT) [40] for end-to-end learning. Consequently, models become highly sensitive to variations in rotational speed, resulting in poor generalization when deployed across different domains. To overcome this limitation, this study incorporates an order transform-based preprocessing method grounded in mechanical principles into the deep learning pipeline.

Although order analysis has also been applied in traditional machine learning-based diagnostic systems, it has typically involved extracting the amplitudes of specific harmonic components (e.g., 1×, 2×, 3×) and feeding them into classifiers such as SVMs or Random Forests [41,42,43]. While effective at summarizing energy in key fault-related frequencies, such approaches inherently discard valuable information present in other spectral regions. They are also limited in their ability to capture latent fault patterns and noise variations, thereby constraining model expressiveness.

In contrast, this study proposes using the entire waveform within the order domain rather than only selected components as the input to the deep learning model. By feeding the full order-domain spectrum to the model, we preserve a wide range of frequency-based characteristics critical for diagnosis, including relative energy distribution across orders, trend shifts, and local patterns. This richer representation enables the model to learn more expressive and discriminative features. The input range is configured to include up to 10 times the fundamental rotational order (10×), thereby maximizing the coverage of physically meaningful frequency bands.

The preprocessing pipeline begins by applying FFT to the input time-series signal. Based on the machine’s rotational speed, the fundamental frequency is calculated and used to convert the frequency axis into the order domain. Because this normalized spectrum aligns fault features along a fixed X-axis regardless of speed, it is well suited for use with the heterogeneous multi-machine, multi-condition dataset used in this study. In other words, by normalizing frequency information across datasets collected from different machines, we enable unified representation learning that is highly beneficial for domain adaptation.

Subsequently, log scaling is applied to control the amplitude’s dynamic range, and MinMax normalization scales the data into the [0, 1] range, making it suitable for neural network input. Finally, a reference signal randomly sampled from normal condition data collected from the same machine is paired with the input after undergoing the same preprocessing steps. This paired data is then used for self-supervised classification based on latent representation differences. The full pseudocode of the preprocessing algorithm is provided in Algorithm 1.
**Algorithm 1** Signal and reference preprocessing pipeline**Require:** Input signal x(t), Rotational speed r^, Machine ID *m***Ensure:** Normalized order-domain pair: (X˜input,X˜ref) 1: **function** Preprocess(x(t),r^) 2:     // Step 1: FFT 3:     X(f)←FFT(x(t)) 4:     // Step 2: Order Transform 5:     f0←r^/60                          ▹ Fundamental frequency 6:     o←f/f0                        ▹ Convert to order domain 7:     Xorder(o)←X(f) 8:     // Step 3: Log Scaling 9:     Xlog(o)←log(1+|Xorder(o)|)10:    // Step 4: Min-Max Normalization11:    X˜(o)←Xlog(o)−min(Xlog)max(Xlog)−min(Xlog)12:    **return** X˜(o)13:**end function**14:**function** FullPipeline(x(t),r^,m)15:    X˜input←PREPROCESS(x(t),r^)16:    // Step 5: Reference Search17:    Randomly select xref(t) from healthy data of machine *m*18:    X˜ref←PREPROCESS(xref(t),r^)19:    **return** (X˜input,X˜ref)20:**end function**

Figure 3 summarizes the overall flow of the preprocessing pipeline. The input raw signal is first transformed into the frequency domain via FFT, then realigned to the order domain using the rotational speed. After scaling and reference pairing, the signal is converted into a unified input format that allows integrated modeling across different machines and operating conditions.

### 3.2. Latent Representation Learning via U-NetVAE with Difference-Aware Classification

The deep learning-based diagnostic model proposed in this study does not simply classify input data. Instead, it quantifies the difference between the input and a normal (reference) signal within an interpretable latent space, and determines anomaly status based on this difference. This framework integrates a VAE structure that learns generalized representations through reconstruction and a classification mechanism that operates on the difference between latent vectors, forming a multi-task learning architecture.

The proposed model, illustrated in Figure 4, consists of three main sub-modules forming a unified architecture: (1) a U-NetVAE encoder that maps the input signal into a latent Gaussian space and captures generalized representations, (2) a decoder that reconstructs the original input from the latent vector to reinforce reconstruction learning, and (3) a classifier that receives the difference between the latent vectors of the input and the reference signal to determine the fault class. This overall pipeline enables both representation learning and classification in a self-supervised manner.

As shown in Figure 5, each encoder and decoder block is constructed using 1D-convolutional residual modules, where skip connections preserve both local and global information throughout the network. The residual blocks include a dimension expansion and contraction process to maximize feature capacity while maintaining sequence structure.

Figure 6 shows the full structure of the U-NetVAE, where the encoder compresses the input sequence and the decoder reconstructs it back to the original shape using transposed convolutions. Skip connections between matching levels ensure feature reuse and reconstruction fidelity.

After feature extraction, the encoder samples from a Gaussian distribution to generate a latent representation z, and incorporates a distribution loss based on KL divergence to ensure the latent space maintains a generalized form. This loss is formulated as in Equation (Equation 1).(1)$$\begin{array}{cc} L_{\mathrm{distribution}} & =\mathrm{KL}(q\left(z|x_{\mathrm{input}}\right)\|p\left(z\right))\\ \; & =\int q\left(z|x_{\mathrm{input}}\right)\log\frac{q(z|x_{\mathrm{input}})}{p(z)}\;dz\\ \; & =\int q\left(z|f_{\mathrm{enc}}\left(x_{\mathrm{input}}\right)\right)\log\frac{q(z|f_{\mathrm{enc}}\left(x_{\mathrm{input}}\right))}{p(z)}\;dz\\ p(z) & ∼N(0,I) \end{array}$$

The decoder mirrors the encoder structure and reconstructs the signal into the original 1D sequence using 1D transpose convolutions. The reconstruction loss is defined using the Huber loss, which provides a balanced trade-off between the mean-squared error (MSE) and mean absolute error (MAE), offering both training stability and robustness to noise. The reconstruction loss is given by Equation (Equation 2).(2)$$L_{\mathrm{recon}}=\left\{\begin{array}{cc} \frac{1}{2}{\left(x_{\mathrm{input}}-x_{\mathrm{recon}}\right)}^{2} & \mathrm{if}\;|x_{\mathrm{input}}-x_{\mathrm{recon}}|\leq \delta \\ \delta \cdot |x_{\mathrm{input}}-x_{\mathrm{recon}}|-\frac{1}{2}\delta^{2} & \mathrm{otherwise} \end{array}\right.$$

The classifier receives the difference vector between $z_{\mathrm{input}}$ and $z_{\mathrm{ref}}$ as the input, and performs classification using a Support Vector Machine (SVM). SVMs are known to perform effectively in margin-based classification tasks in high-dimensional feature spaces, making them well suited for comparison-based decision problems. In this work, the structured nature of the latent difference vector enables reliable anomaly detection using this approach. The classifier is trained using focal loss to handle class imbalance and improve training stability. The focal loss is defined as in Equation (Equation 3).(3)$$L_{\mathrm{classify}}=\left\{\begin{array}{cc} -\alpha_{t}{\left(1-p_{\mathrm{pred}}\right)}^{\gamma }\log\left(p_{\mathrm{pred}}\right) & \mathrm{if}\;y=1\\ -\alpha_{t}{\left(p_{\mathrm{pred}}\right)}^{\gamma }\log\left(1-p_{\mathrm{pred}}\right) & \mathrm{if}\;y=0 \end{array}\right.$$

The overall loss function is formulated as a multi-task objective that includes reconstruction loss, distribution regularization loss, and classification loss. To account for instability in the early stages of training, a beta annealing strategy is employed, in which the weight on the KL-divergence term gradually increases as training progresses. The total loss function is defined as in Equation (Equation 4).(4)$$\begin{array}{cc} L_{\mathrm{total}} & =\beta \left(t\right)\cdot L_{\mathrm{distribution}}+\lambda_{\mathrm{recon}}\cdot L_{\mathrm{recon}}+\lambda_{\mathrm{cls}}\cdot L_{\mathrm{classify}}\\ \beta (t) & =\left\{\begin{array}{cc} \frac{t}{T}\cdot \beta_{\max} & \mathrm{if}\;t\leq T\\ \beta_{\max} & \mathrm{otherwise} \end{array}\right. \end{array}$$

### 3.3. Self-Supervised Test-Time Training for Domain Adaptation

To enable effective adaptation to the target domain in scenarios where source domain data is unavailable, this study adopts a self-supervised learning strategy based on test-time training (TTT). TTT adapts to unlabeled target data by updating only selected modules of a pre-trained model. In our framework, TTT is specifically designed to enhance the representation learning capability of the VAE encoder. The adaptation process consists of two stages, assuming that no labels are available in the target domain after the model has been trained on the source domain.

#### 3.3.1. Step 1: Self-Supervised Encoder Adaptation

For each target domain sample x, two augmented versions are created by adding Gaussian noise at different intensities: a weakly augmented version $x_{\mathrm{weak}}$ and a strongly augmented version $x_{\mathrm{strong}}$. Each of these inputs, along with the original *x*, is passed through the encoder and decoder to obtain the reconstructed outputs $\hat{x}$, ${\hat{x}}_{\mathrm{weak}}$, and ${\hat{x}}_{\mathrm{strong}}$. Based on these, three reconstruction-based losses are defined as shown in Equations (Equation 5)–(Equation 7).

Reconstruction Loss: Loss between original input *x* and reconstruction $\hat{x}$:(5)$$L_{\mathrm{recon}}=\frac{1}{n}\sum_{i=1}^{n}{\left(x_{i}-{\hat{x}}_{i}\right)}^{2}$$Augmentation Loss: Loss between reconstructed weak augmentation $x_{\mathrm{weak}}$ and original input *x*:(6)$$L_{\mathrm{aug}}=\frac{1}{n}\sum_{i=1}^{n}{\left(x_{i}-{\hat{x}}_{\mathrm{weak},i}\right)}^{2}$$Consistency Loss: Loss between reconstructed weak augmentation $x_{\mathrm{weak}}$ and reconstructed strong augmentation $x_{\mathrm{strong}}$:(7)$$L_{\mathrm{consistency}}=\frac{1}{n}\sum_{i=1}^{n}{\left({\hat{x}}_{\mathrm{weak},i}-{\hat{x}}_{\mathrm{strong},i}\right)}^{2}$$

In addition, the KL-divergence loss between the Gaussian latent distributions is computed for each reconstruction and defined as(8)$$L_{\mathrm{KL}}=D_{\mathrm{KL}}^{\mathrm{origin}}+D_{\mathrm{KL}}^{\mathrm{weak}}+D_{\mathrm{KL}}^{\mathrm{strong}}$$

The total loss function is then composed as a weighted sum of the three reconstruction losses and the KL-divergence term, as shown in Equation (Equation 9).(9)$$L_{\mathrm{total}}=L_{\mathrm{recon}}+L_{\mathrm{aug}}+L_{\mathrm{consistency}}+\lambda_{\mathrm{KL}}\cdot L_{\mathrm{KL}},\;\lambda_{\mathrm{KL}}=0.1$$

During this adaptation process, the decoder and classifier are kept frozen, and only the encoder is updated. This design preserves the reconstruction baseline learned from the source domain while allowing the encoder to incorporate structural characteristics specific to the target domain. As a result, the encoder learns a new representation $z^{\prime }$ that is specialized for the target domain while remaining structurally comparable to the original source-based representation.

#### 3.3.2. Step 2: Latent Fusion and Classification

The final classification input is obtained by averaging the latent representation *z* extracted from the encoder before adaptation and the updated representation $z^{\prime }$ obtained after adaptation. This approach effectively integrates the generality of the source domain representation with the specificity of the adapted representation, enabling robust fault diagnosis while mitigating the effects of domain shift in the target domain.

In addition, this averaging can be viewed as a convex combination that explores the intermediate representation in the latent space between the source and target domains, providing an effective feature representation under moderate distribution shifts. This allows the model to retain the general knowledge from the source domain while incorporating instance-specific information from the target domain in a stable and computationally efficient manner.

## 4. Experiment

### 4.1. Dataset Integration

Conventional fault diagnosis models for rotary machinery are typically trained and validated using data collected under a single experimental setup or specific operating conditions. However, such a setup fails to reflect the discrepancies between controlled laboratory environments and the complexities of real-world industrial operations. For instance, vibration signals acquired from laboratory-based machinery are collected under precisely controlled rotational speeds, load conditions, and structural configurations. In contrast, equipment operating in industrial environments is subject to various environmental factors and unpredictable fault scenarios. These differences introduce structural domain shifts that can lead to a significant drop in model performance when a model trained in the laboratory is deployed in the field.

To address this issue, we construct an experimental setup that enables the quantitative evaluation of domain adaptation performance by separating the training and testing datasets across different domains. By explicitly distinguishing between datasets used for training and those used for testing, each reflecting different equipment types and operating conditions, we aim to assess the robustness of fault diagnosis models in settings that simulate real-world deployment scenarios.

To this end, we integrate four publicly available fault diagnosis datasets collected under diverse conditions. Specifically, the Vibration and Motor Current Dataset of Rolling Element Bearing Under Varying Speed Conditions for Fault Diagnosis (VAT), the Mechanical Faults in Rotating Machinery Dataset (DXAI), the VBL-VA001 dataset, and the MaFaulDa dataset are selected. These datasets include major fault conditions commonly found in rotary machinery, making them appropriate for our experimental objectives. Since these datasets differ in terms of their sensor configurations, measurement axes, class definitions, rotational speeds, and fault scenarios, we standardize them into a unified input format and class structure. This preprocessing ensures compatibility for cross-domain validation experiments and enables the evaluation of adaptation across heterogeneous environments. Table 1 and Table 2 describe the environments and other information for each dataset.

#### 4.1.1. VAT

The VAT dataset (Figure 7), was collected by the Korea Advanced Institute of Science and Technology (KAIST) and contains multi-sensor data—including vibration, current, acoustic, and temperature signals measured from rotary machinery under various load and speed conditions. The dataset consists of two configurations. The first includes measurements under three different load conditions: 0 Nm, 2 Nm, and 4 Nm. The second includes bearing fault data (inner race, outer race, and ball faults) recorded at varying rotational speeds ranging from 680 RPM to 2460 RPM.

#### 4.1.2. DXAI

The DXAI dataset (Figure 8), is a vibration dataset constructed by the Universidade Federal de São João del-Rei in Brazil. It includes four operating conditions: normal, unbalance, misalignment, and looseness. The data were collected under experimental conditions designed to closely resemble real industrial environments. The test bench consists of components such as a motor, an inverter, bearings, and pulleys. Each fault condition was repeated five times, with a randomized experiment order and reinitialization to ensure data diversity.

#### 4.1.3. VBL-VA001

The VBL-VA001 dataset (Figure 9), was collected by the Sepuluh Nopember Institute of Technology in Indonesia. It contains data measured at a fixed rotational speed of 3000 RPM and includes four fault conditions: normal, misalignment, unbalance (at two severity levels), and BPFO (Ball Pass Frequency Outer race) faults. The dataset consists of four classes, each with 1000 samples. All experiments were conducted at a sampling frequency of 20 kHz.

#### 4.1.4. MaFaulDa

The MaFaulDa dataset (Figure 10), was developed by the Federal University of Rio de Janeiro in Brazil, using the Alignment–Balance–Vibration Trainer (ABVT) to simulate various fault conditions. The dataset includes several classes: normal, misalignment (horizontal and vertical), unbalance, Ball Pass Frequency Outer race (BPFO), Ball Pass Frequency Inner race (BPFI), and bearing cage faults. It also incorporates variations in fault severity and rotational speeds ranging from 737 RPM to 3686 RPM. All samples were recorded at a 50 kHz sampling rate for a duration of 5 s.

#### 4.1.5. Integration Setting and Results

Since the original datasets differ in terms of class definitions and measurement conditions, we refined them into a unified and interpretable class structure. In particular, the MaFaulDa dataset includes various speeds and load levels even within the same fault category (e.g., unbalance), and misalignment and looseness conditions are collected under horizontal/vertical and different fastening levels, respectively. To avoid excessive subclass fragmentation, we integrated the data into five primary fault categories: normal, misalignment, unbalance, and bearing. The bearing class merges BPFO, BPFI, and bearing cage-related faults. The final class definitions are as follows:Normal: healthy operating condition.Misalignment: shaft misalignment.Unbalance: rotor mass imbalance.Bearing: bearing-related faults (including outer race, inner race, and cage faults).

In rotary machinery diagnostics, orthogonal sensor axes (X and Y) are typically used together. In this study, we utilized both X-axis and Y-axis vibration signals measured from sensors mounted on the motor. The final integrated dataset is as shown in Table 1 and Table 2. The total dataset consists of 62,863 samples. To address the known issue of test–train leakage in condition-based maintenance studies, especially those relying on single-source datasets like CWRU, we constructed a benchmark that integrates four diverse public datasets. These datasets were collected under different machines, sensor setups, rotational speeds, and acquisition protocols. This diversity reduces the risk of model overfitting to a narrow fault distribution.

Moreover, our experimental design strictly separates the source and target domains. VAT, MaFaulDA, and DXAI are used exclusively for training and validation, while the VBL-VA001 dataset is reserved as a test-only target domain. No data from the target domain was used during training, which structurally prevented information leakage and ensured a fair evaluation under realistic domain shift conditions. Table 3 summarizes the integrated dataset distribution. The training–validation–test split was configured as follows: VBL-VA001, MaFaulDa, and DXAI were used as source domains, with an 8:2 split for training and validation, respectively. The VAT dataset was designated as the target domain and was excluded from training. It was used exclusively as a test set to evaluate model adaptation under domain shift conditions.

### 4.2. Benchmark

#### 4.2.1. Benchmark Setting

To evaluate the performance of the proposed model, we conducted benchmark experiments against a variety of conventional models. The comparison included both machine learning-based and deep learning-based methods. For machine learning models, we used the Support Vector Machine (SVM) [48], Logistic Regression [49], Random Forest [50], Gradient Boosting [51], and K-Nearest Neighbors (KNN) [52] methods. For deep learning models, we adopted 1D-Convolutional Neural Network (1D-CNN) [53], Long Short-Term Memor y(LSTM) [54], and Transformer [55] architectures.

For all machine learning models, the hyperparameters were optimized using grid search on the source domain training data. The deep learning models were implemented with the following architectural settings:1D-CNN: This is composed of three 1D convolutional blocks followed by a fully connected layer. Each block consists of a convolution layer, batch normalization, ReLU activation, and max pooling. The convolution layers use a kernel size of 3, a stride of 1, and padding of 1, with the output channels set to 16, 32, and 64, respectively.LSTM: This is constructed with two LSTM layers, each with a hidden dimension of 128. A dropout rate of 0.3 is applied between layers, and the input size is set to 2.Transformer: The embedding dimension is set to 64, and the number of heads in the multi-head attention is 4. The encoder output is summarized and passed through a fully connected layer to predict fault classes.

All deep learning models were trained using CrossEntropyLoss and the Adam optimizer, with a learning rate of $1\times 10^{-3}$, a dropout rate of 0.3, and a batch size of 512, over 200 epochs. To assess the effectiveness of the proposed order-frequency-based preprocessing technique, we applied five different input data types across all models:Statistical features from the frequency domain: power, max frequency, mean frequency, median frequency, spectral skewness, spectral kurtosis, peak amplitude, band energy, dominant frequency power, spectral entropy, root-mean-square (RMS) frequency, frequency variance.Statistical features from the time domain: mean, standard deviation, max, min, RMS, skewness, kurtosis, peak, peak-to-peak value, crest factor, impulse factor, shape factor.Raw time-series signals.Frequency-domain signals generated through FFT transformation.Order-frequency-domain signals produced by the proposed preprocessing pipeline.

For machine learning models, the first two statistical feature sets, (1) and (2), were used as inputs. For deep learning models, raw signals (3), FFT-based inputs (4), and order-frequency-transformed inputs (5) were used. These results empirically validate the effectiveness of the proposed preprocessing method, and provide a comprehensive evaluation of the effectiveness of the proposed input transformation.

#### 4.2.2. Benchmark Results

To evaluate the effectiveness of the proposed cross-domain fault classification model, we conducted benchmark experiments comparing it with various existing techniques. The models used in the experiments include traditional machine learning approaches and deep learning-based architectures. Each model was trained using either raw vibration signals or transformed inputs, allowing for a comparison of performance across different modeling strategies.

All models were trained using data collected from the source domain. Performance evaluation was carried out on two datasets: a validation set from the source domain (i.e., without domain shift) and a test set from the target domain (i.e., with domain shift). The evaluation metrics used were precision, recall, and F1-score. These three metrics collectively assess the model’s accuracy and sensitivity, and the trade-off between them. The benchmark results are shown in Table 4.

Precision measures the proportion of correctly predicted positive samples among all samples predicted as positive. A higher precision indicates a lower false positive rate.(10)$$\mathrm{Precision}=\frac{TP}{TP+FP}$$

Recall measures the proportion of true positive samples correctly identified by the model. A higher recall indicates better sensitivity to actual positives.(11)$$\mathrm{Recall}=\frac{TP}{TP+FN}$$

The F1-score is the harmonic mean of precision and recall, providing a balanced evaluation when both metrics are equally important. It yields a high value only when both precision and recall are sufficiently high.(12)$$F1=2\times \frac{Precision\times Recall}{Precision+Recall}$$

In the analysis of the experimental results, we present visual comparisons of the F1-score and recall across different models and preprocessing methods. The F1-score, as the harmonic mean of precision and recall, provides a comprehensive measure of overall model performance, while recall is particularly sensitive to performance degradation caused by domain shift. Thus, these two metrics are primarily used in this analysis. In the source domain, most models achieved high performance, with precision, recall, and F1-score all exceeding 0.9. However, under domain shift, performance varied significantly depending on the model type.

As shown in Figure 11, machine learning models experienced a sharp decline in performance. On average, their F1-score dropped from above 0.9 to below 0.1, and recall decreased dramatically to the 0.05 0.25 range. For example, SVM achieved an F1-score of 0.92 in the source domain but only 0.10 in the target domain, corresponding to an 89% performance drop.

In contrast, as shown in Figure 12, deep learning models demonstrated more stable adaptation performance. While 1D-CNN and LSTM still suffered from degradation, they maintained average F1-scores in the 0.2–0.3 range in the target domain, indicating greater robustness than traditional machine learning models.

The full framework proposed in this study, which integrates both the preprocessing pipeline and model architecture, demonstrated consistent and superior performance. It achieved a precision of 0.89, recall of 0.84, and an F1-score of 0.84 in the source domain, and maintained strong target-domain results with a precision of 0.50, recall of 0.51, and an F1-score of 0.47. Importantly, while most baseline models recorded target-domain recall in the range of 0.1–0.3, our model consistently outperformed them, achieving a recall of 0.51 and demonstrating enhanced anomaly detection sensitivity and domain adaptation capability.

### 4.3. Model Implementation and Ablation Study

To ensure stable performance under domain shift, this study introduces three key components: (1) A preprocessing method based on order-frequency transformation. (2) Self-supervised reconstruction learning using a variational autoencoder (VAE). (3) A test-time training (TTT) strategy for unsupervised adaptation to the target domain. To quantitatively assess the contribution of each component, we conduct an ablation study.

#### 4.3.1. Model Implementation

The proposed U-NetVAE model is a U-Net-based variational autoencoder designed for 1D time-series data. The encoder–decoder architecture is enhanced with skip connections to improve reconstruction performance. The latent vectors extracted from the encoder are passed to a downstream classifier, which determines the fault class based on the difference between the latent vectors of the input and a normal (reference) signal. Although our framework is conceptually inspired by SVM-based classification in latent space, we implement it using a single fully connected layer that performs the same linear decision function. This enables end-to-end training with the encoder.

The architecture is configured as follows: the U-NetVAE consists of approximately 117 K parameters, and the classifier has 16.4 K parameters, resulting in a total of about 133 K parameters. The main hyperparameters for the U-NetVAE are as follows: 2 residual blocks, each with 8 residual layers; a latent embedding dimension of 32; and an embedding length of 128. The model is trained with a batch size of 512 for 400 epochs. StepLR is used as the learning rate scheduler for the U-NetVAE, while the classifier uses a Cosine Annealing scheduler.

To prevent early overfitting in the classifier, a 50-epoch warm-up strategy is applied. Both the U-NetVAE and classifier are optimized using the Adam optimizer with an initial learning rate of $1\times 10^{-3}$. To ensure reproducibility, all experiments are conducted with the random seed fixed at 42. The overall training structure is summarized in Table 5.

The test-time training (TTT) strategy is implemented in an offline manner, where only the encoder is updated. TTT leverages unlabeled target domain data to refine the latent representations via self-supervised learning. During this phase, the decoder and classifier are kept frozen. The TTT hyperparameters are as follows: a batch size of 512, a learning rate of $1\times 10^{-5}$, 2 training epochs, and optimization using SGD. This structure is designed to provide effective domain adaptation with minimal computational overhead.

#### 4.3.2. Ablation Study

To quantitatively analyze the contribution of each component in the proposed framework, we conducted an ablation study. The experiments were performed using four different model configurations, as summarized in the following Table 6.

The Baseline model consists only of the encoder, and resembles a 1D-CNN architecture with residual connections. Order-frequency preprocessing, reconstruction-based learning, and TTT strategies are all disabled in this configuration. While the model showed stable performance in the source domain, its performance dropped sharply in the target domain, with an F1-score of just 0.23, indicating poor adaptation to domain shifts.

When order-frequency preprocessing was added to the baseline, performance in the target domain improved significantly: precision improved from 0.26 to 0.48, recall by 0.24 points, and the F1-score by 0.23 points. This result highlights the effectiveness of the proposed preprocessing method in mitigating cross-domain distributional shifts and extracting robust features. In the next configuration, reconstruction learning was introduced as an auxiliary task. Although no notable performance gains were observed in either domain, the integration of the VAE structure enabled latent adaptation during the TTT phase. Thus, while the numerical metrics remained similar, the model gained the flexibility to adapt in a self-supervised setting, enhancing its extensibility.

Finally, applying TTT to the full proposed framework led to further improvements in the target domain: precision increased by 0.02 points, recall by 0.01 points, and the F1-score by 0.04 points. This demonstrates that even with short adaptation periods, TTT can compensate for residual performance drops due to domain shift. The increase in F1-score reflects overall performance gains, and the stable recall indicates reliable anomaly detection capability. In summary, each component contributed to performance improvement on its own, and the integrated full framework played a critical role in achieving robust adaptation to unseen target domains.

To further assess the impact of the reconstruction module architecture, we conducted a comparative experiment using three variants: a standard VAE, a vector-quantized VAE (VQ-VAE), and the proposed U-NetVAE. All models were implemented with the same number of layers, latent dimensionality, and optimization settings to ensure fair comparison. Test-time training (TTT) was applied to each model before evaluation.

As shown in Table 7, in the source domain, all models performed relatively well. However in the target domain, the U-Net-VAE showed a notable improvement in the F1-score (0.50) compared to the VQ-VAE (0.35) and standard VAE (0.15). These results indicate that the U-Net architecture plays a key role in enhancing generalization under domain shift by enabling more robust latent representation learning.

According to prior studies [56], updating only the encoder during test-time training while jointly performing a self-supervised task has been reported to be effective. Based on these findings, we adopted the same approach in our work, and as a result, achieved stable performance, with a recall of 0.51, precision of 0.50, and an F1-score of 0.50 when updating only the encoder. Additionally, when we included the classifier in the test-time training along with the encoder, we obtained a recall of 0.48, precision of 0.46, and an F1-score of 0.46.

Through these comparative experiments, we confirmed that updating only the encoder provides more stable and generalized performance and can be reliably utilized under domain shift conditions. In contrast, including the classifier in the adaptation process can lead to partial performance improvements; however, it was observed that this approach may result in unstable convergence and potential overfitting depending on the distributional differences between domains.

Additionally, to further evaluate the effectiveness of different test-time training (TTT) methods under identical conditions, we conducted comparative experiments using the same dataset with T3A [57] and TAST [58], which are methods used in time-series data [59,60], and the proposed U-Net-VAE-based TTT approach. As summarized in the Table 8, in the target domain, T3A achieved a precision of 0.28, recall of 0.33, and an F1-score of 0.30, while TAST achieved a precision of 0.43, recall of 0.31, and an F1-score of 0.36. In contrast, the proposed U-Net-VAE-based TTT method achieved the highest performance with a precision of 0.50, recall of 0.51, and an F1-score of 0.50. These findings confirm that our U-Net-VAE-based TTT framework not only achieves superior adaptation performance under domain shifts but also maintains a general and practical architecture that is readily applicable to industrial time-series fault diagnosis without requiring additional architectural modifications.

## 5. Conclusions

In this study, we proposed a robust fault diagnosis framework for rotary machinery under domain shift conditions. Our method integrates order-frequency-domain preprocessing, self-supervised latent representation learning with a U-NetVAE, and a source-free test-time training (TTT) strategy. Together, these components enable effective domain adaptation without access to source data or target labels. Through extensive benchmark experiments across four public datasets, the proposed approach consistently outperformed conventional models in the presence of domain shift. Ablation studies confirmed the individual contributions of each component.

This work demonstrates the effectiveness of combining mechanical priors, self-supervised learning, and lightweight adaptation for practical industrial diagnostics. Future research will focus on extending the model to handle open-set scenarios and unseen fault types, as well as improving its applicability to low-resource or online monitoring environments.

## Figures and Tables

**Figure 1 sensors-25-04383-f001:**
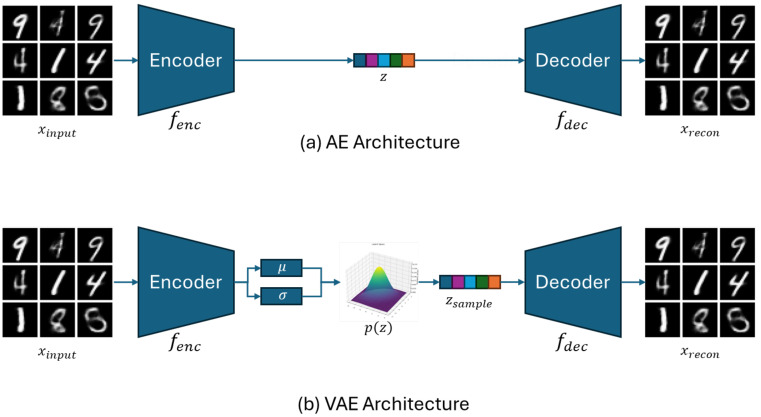
Autoencoder (AE) and variational autoencoder (VAE) architecture. (**a**) In the AE, the encoder $f_{\mathrm{enc}}$ maps the input $x_{\mathrm{input}}$ to a latent vector *z*, and the decoder $f_{\mathrm{dec}}$ reconstructs $x_{\mathrm{recon}}$ from *z*. (**b**) In the VAE, the encoder $f_{\mathrm{enc}}$ computes the mean vector μ and standard deviation vector σ of the latent-space distribution from the input $x_{\mathrm{input}}$. A sample $z_{\mathrm{sample}}$ is then drawn from the normal distribution $p\left(z\right)=N\left(\mu ,\sigma^{2}\right)$ and passed to the decoder $f_{\mathrm{dec}}$ to reconstruct $x_{\mathrm{recon}}$.

**Figure 2 sensors-25-04383-f002:**
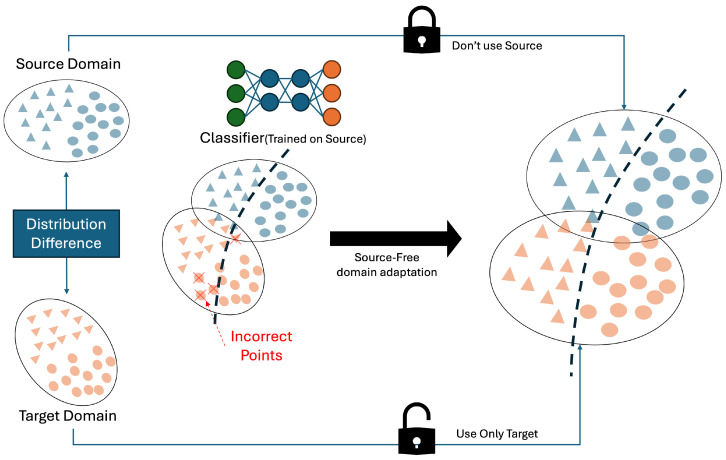
Source-Free Domain Adaptation Framework. Blue indicates source-domain samples orange indicates target-domain samples. The “Distribution Difference” arrow highlights the discrepancy between the two domains. The classifier is trained on source data, and target samples misclassified in the mixed set are marked with red crosses X. The closed lock icon signifies that source data cannot be used during adaptation, while the open lock icon indicates that only target data is used. The dashed line denotes the decision boundary of the adapted classifier.

**Figure 3 sensors-25-04383-f003:**
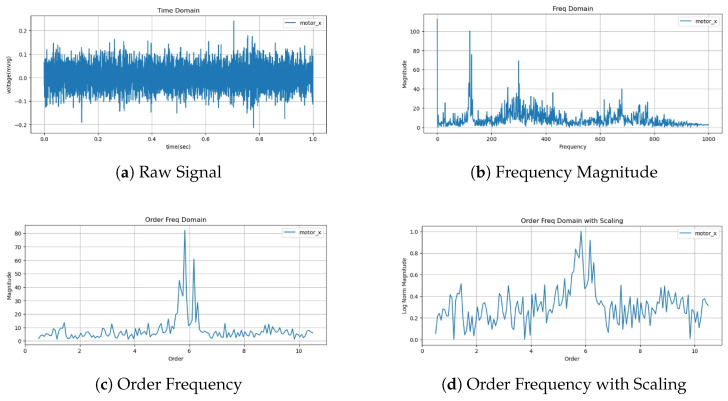
Data preprocessing results.

**Figure 4 sensors-25-04383-f004:**
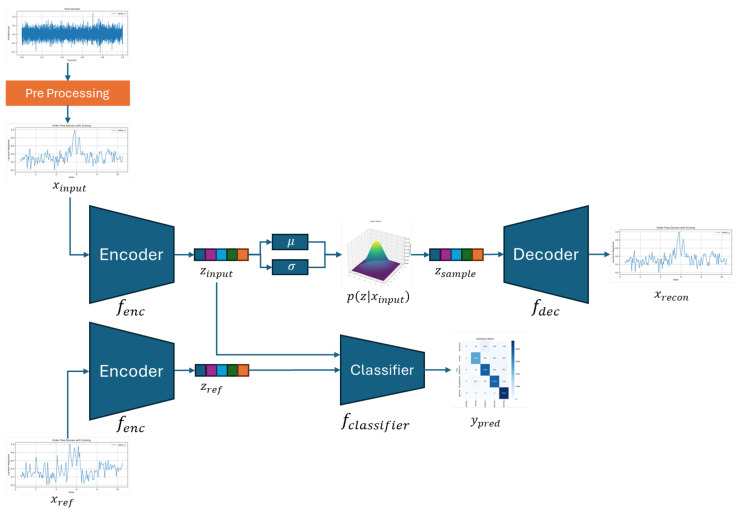
Proposed method.

**Figure 5 sensors-25-04383-f005:**
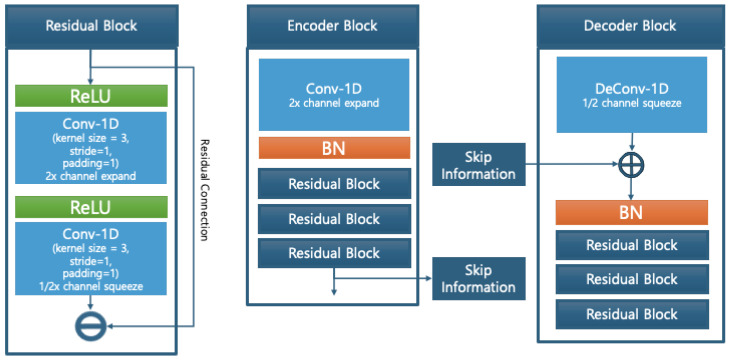
Model block structure.

**Figure 6 sensors-25-04383-f006:**
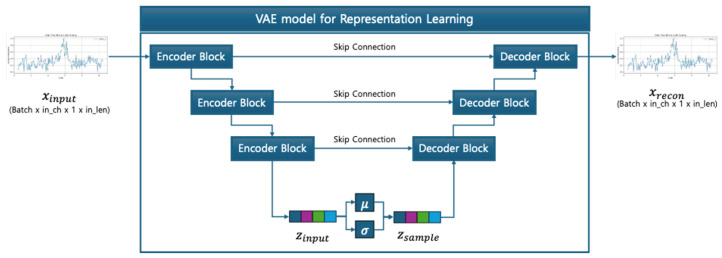
U-Net-based VAE structure.

**Figure 7 sensors-25-04383-f007:**
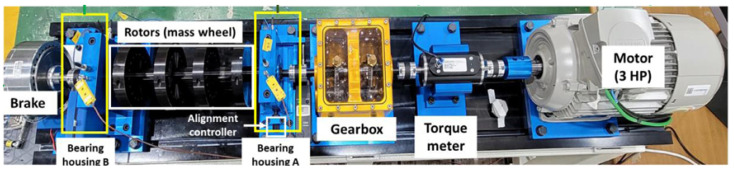
VAT data collection environment.

**Figure 8 sensors-25-04383-f008:**
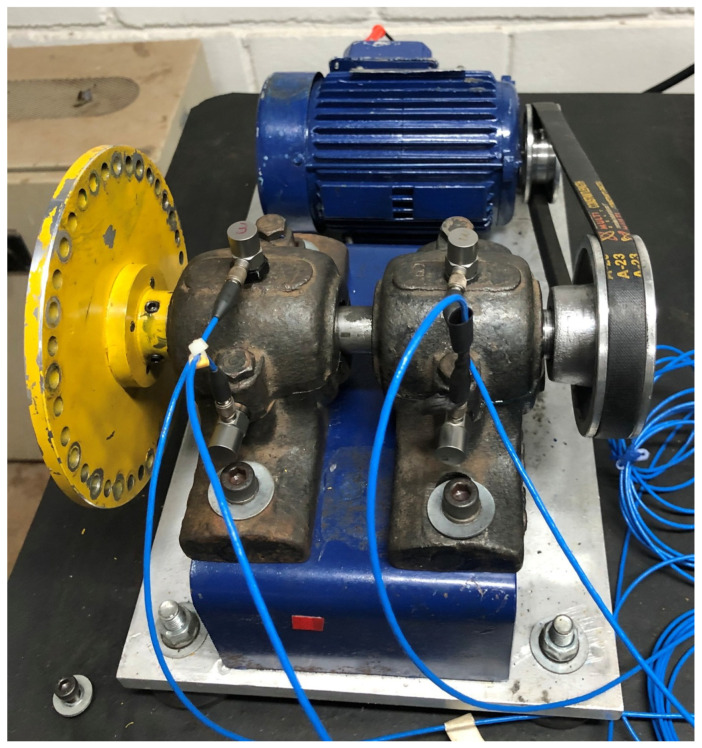
DXAI data collection environment.

**Figure 9 sensors-25-04383-f009:**
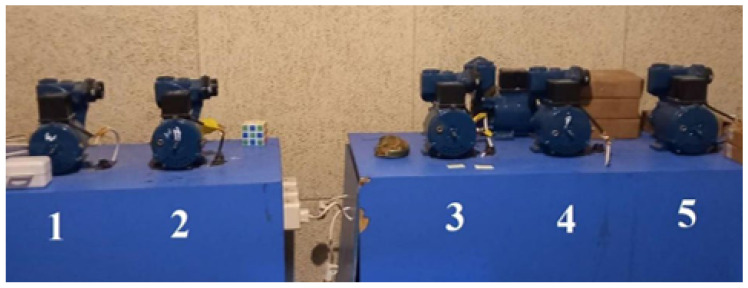
VBL data collection environment.

**Figure 10 sensors-25-04383-f010:**
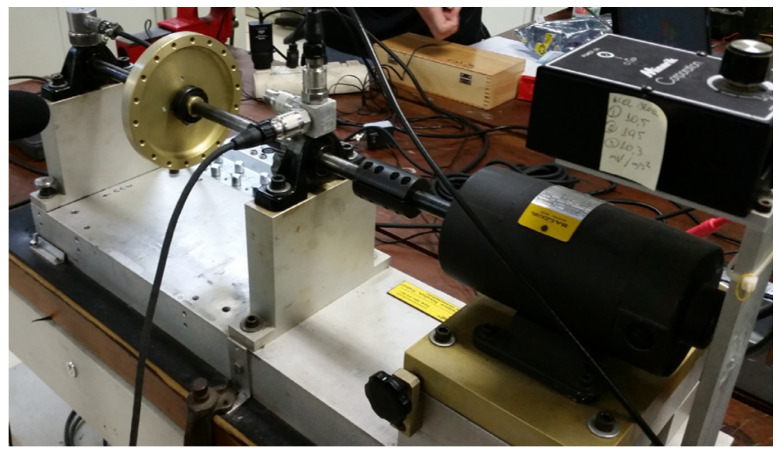
MaFaulDa data collection environment.

**Figure 11 sensors-25-04383-f011:**
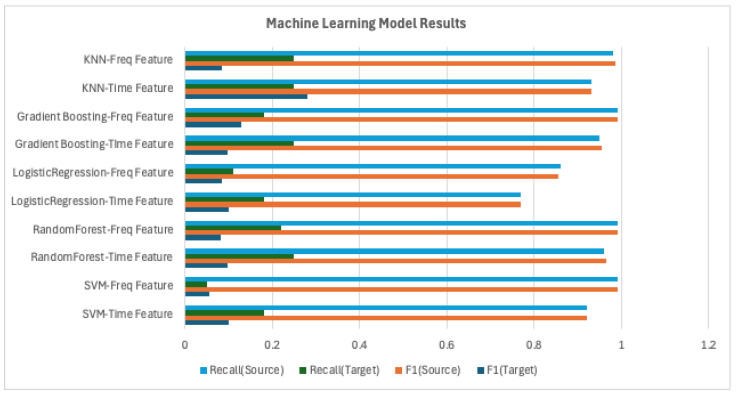
Machine learning model results.

**Figure 12 sensors-25-04383-f012:**
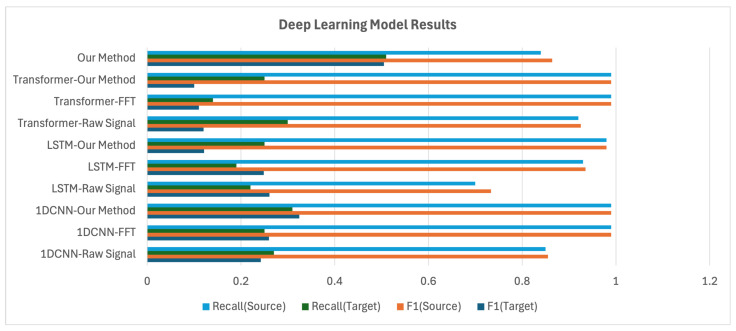
Deep learning model results.

**Table 1 sensors-25-04383-t001:** Dataset specifics (Part 1: VAT and VBL-VA001).

	Datasets	VAT	VBL-VA001
Items			
Motor		Four-pole AC motor by SIEMENS, Germany (3HP)	Panasonic, Japan GP-129JXK (125W, 3000RPM)
Accelerometers		4× PCB352C34 at bearings A, B, United States	LOG-0002-100G on motor (X,Y,Z), United States
Other Sensors		Acoustic mic, 4× CT sensor, 2× K-type thermocouples	-
Sampling Frequency		25.6 kHz (vibration/temp), 51.2 kHz (acoustic)	20 kHz
Fault Types		Bearing fault, unbalance, misalignment	Normal, misalignment, unbalance (2 levels), BPFO
Operating Conditions		Load: 0, 2, 4 Nm/RPM: 3010, 680–2460	Fixed 3000 RPM
Acquisition Time and Repetition		Normal: 120s, Faulty: 60 s, 7 × 300 s, 1 × 600 s	1000 samples/class, 5 s/sample
Data Collection Institution		Korea Advanced Institute of Science and Technology	Sepuluh Nopember Institute of Technology, Indonesia
Source		Jung et al. (2023) [44], Mendeley Data, 10.17632/vxkj334rzv.7	Atmaja et al. (2024) [45], J. Vibration Eng. Tech. 12(2)

**Table 2 sensors-25-04383-t002:** Dataset specifics (Part 2: MaFaulDA and DXAI).

	MaFaulDA	DXAI
Motor	DC motor, 1/4CV, ABVT trainer, United States	Three-phase induction motor, B56 B4, 1650 RPM, 0.09 kW, 220 V, 0.70 A, Germany
Accelerometers	IMI 601A01, IMI 604B31, United States	4× PCB352C33 (2 per bearing), United States
Other Sensors	Shure(United States) SM81 condenser microphone	-
Sampling Frequency	50 kHz	25 kHz
Fault Types	8568 labeled faults	Normal, unbalance, misalignment, looseness
Operating Conditions	60–3686 RPM (varied)	Fixed 1238 RPM
Acquisition Time and Repetition	1951 samples (varied), 5 s/sample	5 tests/class, 420 samples/test, 1 s/sample
Data Collection Institution	Federal University of Rio de Janeiro	Univ. of São João del-Rei, Universidade Federal de Uberlandia, Universita degli Studi di Padova
Source	MaFaulDa Database (2018) [46], www02.smt.ufrj.br/~offshore/mfs, accessed on 17 April 2025	Lucas Brito et al. (2022) [47], Expert Systems with Applications, 10.1016/j.eswa.2023.120860

**Table 3 sensors-25-04383-t003:** Data integration results.

	Dataset	VAT-MCD	VBL-VA001	MaFaulDA	DXAI	Total
Class						
normal		1077	8000	392	2100	11,569
misalignment		2151	8000	3984	2100	16,235
unbalance		3585	8000	2664	2100	16,349
bearing		2142	8000	8568	-	18,710
Total		8955	32,000	15,608	7145	62,863

**Table 4 sensors-25-04383-t004:** Experimental results comparison for domain adaptation.

Model Name	Data Preprocessing Method	Validation (Source Domain)	Test (Target Domain)
SVM	Time Feature	precision: 0.92; recall: 0.92; F1: 0.92	precision: 0.07; recall: 0.18; F1: 0.09
	Frequency Feature	precision: 0.99; recall: 0.99; F1: 0.99	precision: 0.06; recall: 0.05; F1: 0.05
Random Forest	Time Feature	precision: 0.97; recall: 0.96; F1: 0.97	precision: 0.06; recall: 0.25; F1: 0.10
	Frequency Feature	precision: 0.99; recall: 0.99; F1: 0.99	precision: 0.05; recall: 0.22; F1: 0.08
Logistic Regression	Time Feature	precision: 0.77; recall: 0.77; F1: 0.76	precision: 0.07; recall: 0.18; F1: 0.10
	Frequency Feature	precision: 0.85 recall: 0.86;; F1: 0.85	precision: 0.07; recall: 0.11; F1: 0.08
Gradient Boosting	Time Feature	precision: 0.96; recall: 0.95; F1: 0.96	precision: 0.06; recall: 0.25; F1: 0.10
	Frequency Feature	precision: 0.99; recall: 0.99; F1: 0.99	precision: 0.10; recall: 0.18; F1: 0.12
KNN	Time Feature	precision: 0.93; recall: 0.93; F1: 0.93	precision: 0.32; recall: 0.25; F1: 0.28
	Frequency Feature	precision: 0.99; recall: 0.98; F1: 0.99	precision: 0.05; recall: 0.25; F1: 0.08
1D-CNN	Raw Signal	precision: 0.86; recall: 0.85; F1: 0.85	precision: 0.22; recall: 0.27; F1: 0.24
	FFT	precision: 0.99; recall: 0.99; F1: 0.99	precision: 0.27; recall: 0.25; F1: 0.25
	Our Method	precision: 0.99; recall: 0.99; F1: 0.99	precision: 0.34; recall: 0.31; F1: 0.32
	Raw Signal	precision: 0.77; recall: 0.70; F1: 0.69	precision: 0.32; recall: 0.22; F1: 0.26
LSTM	FFT	precision: 0.94; recall: 0.93; F1: 0.93	precision: 0.36; recall: 0.19; F1: 0.24
	Our Method	precision: 0.98; recall: 0.98; F1: 0.98	precision: 0.08; recall: 0.25; F1: 0.12
	Raw Signal	precision: 0.93; recall: 0.92; F1: 0.92	precision: 0.40; recall: 0.30; F1: 0.21
Transformer	FFT	precision: 0.99; recall: 0.99; F1: 0.99	precision: 0.34; recall: 0.14; F1: 0.11
	Our Method	precision: 0.99; recall: 0.99; F1: 0.99	precision: 0.57; recall: 0.25; F1: 0.10
Our Method	Our Method	precision: 0.89; recall: 0.84; F1: 0.86	precision: 0.50; recall: 0.51; F1: 0.50

**Table 5 sensors-25-04383-t005:** Training setting.

Model	Scheduler	Optimizer (Training)	Optimizer (TTT)
U-NetVAE	StepLR	Adam	SGD (Only Encoder)
Classifier	Cosine Annealing	Adam	-

**Table 6 sensors-25-04383-t006:** Ablation study results.

Experiment Setting	Order Preprocessing	Reconstruction	TTT	Source Domain Test	Target Domain Test
Baseline	x	x	x	precision: 0.89 recall: 0.83 F1: 0.82	precision: 0.26 recall: 0.26 F1: 0.23
+Order Spectrum Preprocessing	o	x	x	precision: 0.89 recall: 0.84 F1: 0.84	precision: 0.48 recall: 0.50 F1: 0.46
+Reconstruction	o	o	x	precision: 0.89 recall: 0.84 F1: 0.84	precision: 0.48 recall: 0.50 F1: 0.46
+Ours	o	o	o	precision: 0.89 recall: 0.84 F1: 0.86	precision: 0.50 recall: 0.51 F1: 0.50

Legend: o = included; x = not included.

**Table 7 sensors-25-04383-t007:** Comparison of classification performance using different reconstruction modules after applying test-time training (TTT).

Model	Precision (Source)	Recall (Source)	F1 (Source)	Precision (Target)	Recall (Target)	F1 (Target)
VAE	0.86	0.74	0.80	0.15	0.16	0.15
VQ-VAE	0.81	0.78	0.79	0.38	0.32	0.35
U-Net-VAE (Ours)	**0.89**	**0.84**	**0.86**	**0.50**	**0.51**	**0.50**

**Table 8 sensors-25-04383-t008:** Comparison of classification performance by applying different test-time training (TTT).

Model	Precision (Target)	Recall (Target)	F1 (Target)
T3A	0.28	0.33	0.30
TAST	0.43	0.31	0.36
U-Net-VAE (Ours)	**0.50**	**0.51**	**0.50**

## Data Availability

No new data were created or analyzed in this study.

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
