# Peer review of "Source-Free Domain Adaptation Framework for Rotary Machine Fault Diagnosis"

_sensors, 2025, doi:10.3390/s25144383_

Round 1

Reviewer 1 Report

Comments and Suggestions for Authors In this manuscript, the authors propose the "Source Free Domain Adaptation Framework for Rotary Machine Fault Diagnosis." Overall, the structure of this manuscript is relatively complete, but there are some serious issues that need to be addressed. The specific comments are as follows:
  1. The contributions stated in the introduction are inappropriate. For example, there is no essential difference between contribution 1 and contribution 2.
  2. The introduction lacks necessary references.
  3. Why did the authors choose to use frequency data as the input for the model? Most of the related works use raw signals as the input for the model, and the authors should explain the motivation behind this choice.
  4. The authors should provide a more detailed comparison with existing domain adaptation techniques, especially those specifically tailored for time-series data.
  5. The authors should consider conducting additional ablation studies to explore the impact of different network architectures on the model's performance. This would provide deeper insights into the design choices.
  6. There are serious formatting issues in the manuscript, especially in the experimental section. For example, some sections lack numbering.
  7. The figures in the manuscript need more explanation and description. In addition, some figures are quite inappropriate, such as Figure 7.
  8. The conclusion is too long, and the experimental results in the conclusion should be shortened.
  9. There are numerous textual errors in the manuscript. For example, "Adapatation" in the title should be "Adaptation".

Author Response

Comment1 : The contributions stated in the introduction are inappropriate. For example, there is no essential difference between contribution 1 and contribution 2.

Response1 : Thank you for pointing this out. We agree that the original contributions (1) and (2) overlapped in their description of the reconstruction-based framework. To address this, we revised the contribution section by restructuring it into three clearly distinct points. Please refer to the updated paragraph at the end of the Introduction (Page 2).

Comment2 : The introduction lacks necessary references.

Response2 : Thank you for the valuable comment. We agree that our Introduction lacked sufficient citations to support key statements. In the revised manuscript, we have carefully reviewed the related literature and added references to strengthen the background and motivation. These additions can be found in Introduction (Page 1–2).

Comment3 : Why did the authors choose to use frequency data as the input for the model? Most of the related works use raw signals as the input for the model, and the authors should explain the motivation behind this choice.

Response3 : Thank you for this insightful comment. We agree that many previous studies use raw time-domain vibration signals as input. However, vibration signals are inherently non-stationary and strongly affected by varying speeds and loads, which can make raw inputs less consistent across domains. To justify our choice of frequency-domain inputs, we added a paragraph in Section 2.1 that reviews prior work comparing time, frequency, and order domain representations. This choice is supported both theoretically, based on domain-specific literature, and experimentally, as demonstrated by the input-type comparison results in Table 4.

Comment4 : The authors should provide a more detailed comparison with existing domain adaptation techniques, especially those specifically tailored for time-series data.

Response4 : Thank you for your valuable comment. We agree that providing a detailed comparison with existing domain adaptation techniques will enhance the clarity and contribution of our work. Most existing SFDA methods have mainly been explored in the image domain, making it challenging to apply them to time-series data. To address this, prior studies have applied T3A and TAST to EEG signal analysis and fault detection tasks, and in this study, we implemented and evaluated these methods. The results are included in the last paragraph of Section 4.3.2 of the revised manuscript.

Comment5 : The authors should consider conducting additional ablation studies to explore the impact of different network architectures on the model's performance. This would provide deeper insights into the design choices.

Response5 : Thank you for your suggestion. To address this, we conducted an additional ablation study comparing three different architectures for the reconstruction module: standard VAE, VQ-VAE, and the proposed UNet-VAE. All models were trained under the same experimental settings and tested after applying test-time training (TTT). We evaluated their classification performance in both the source and target domains using precision, recall, and F1-score. The results, summarized in Table 7, show that the UNet-VAE consistently outperformed the alternatives, especially in the target domain. These additions have been reflected in Section 4.3.2 of the revised manuscript.

Comment6 : There are serious formatting issues in the manuscript, especially in the experimental section. For example, some sections lack numbering.

Response6 : Thank you for your helpful observation. In the original version, some parts of the experimental section were presented as standalone paragraphs without subsection numbering, which may have caused confusion. Following your suggestion, we revised the formatting to explicitly include subsection numbers (e.g., 4.2.1 Benchmark Setting, 4.2.2 Benchmark Result) to improve clarity and consistency throughout the experimental section.

Comment7 : The figures in the manuscript need more explanation and description. In addition, some figures are quite inappropriate, such as Figure 7.

Response7 : Thank you for the helpful comment. We agree that the initial descriptions of key figures, particularly Figures 4, 5, and 6, were insufficient and may have appeared disconnected from the main text. To address this, we revised Section 3.2 to integrate the explanation of each figure directly into the surrounding paragraphs. These changes improve the clarity and relevance of each figure within the context of the model design. As for Figure 7, we acknowledge that the original image suffered from low resolution and visual clarity. In the revised version, we have replaced it with a high-resolution version to improve its readability and presentation quality.

Comment8 : The conclusion is too long, and the experimental results in the conclusion should be shortened.

Response8 : Thank you for your valuable feedback. We agree that our conclusion was unnecessarily long and included repetitive details from the experimental section. To address this, we have revised the conclusion to be more concise by focusing on the overall contributions and implications of the study. Detailed numerical results have been removed, and the text now emphasizes the high-level findings and future directions. These can be found in Conclusion

Comment9 : There are numerous textual errors in the manuscript. For example, "Adapatation" in the title should be "Adaptation".

Response9 : Thank you for pointing this out. We sincerely apologize for the typographical error in the title. We have corrected “Adapatation” to “Adaptation” and carefully proofread the entire manuscript to fix additional typographical and grammatical issues throughout the text. These revisions have been reflected in the updated version.

Reviewer 2 Report

Comments and Suggestions for Authors

This paper deals with fault diagnosis in rotating machinery. It focuses on the problem of domain shift conditions. As far as I understand (maybe I wron here) they try to solve the problem of transfer across different operating condition, which is known as transfer in the identical machine (TIM). The algorithm they presented combine preprocessing based on order-frequency, variational autoencoder for representatin learning and test-time training for source-free domain adaptation. The paper demonstrated the algorithm using four public availebel datasets. They conpare their results with other approches. Furthemore, the paper investigate the contribution of each component in the algorithm.

I have the following questions/comments on the manuscripts:

  1. There is a known problem of test-training leakage in the field of condition based maintenance by vibration analysis because the number of different types of faults in public available dataset is very low (e.g., CRWU contains less than 20 different real fault shapes). Why this paper does not have this problem?
  2. The paper does not include literature background on other approaches of zero-fault learning such as “Zero-fault-shot learning for bearing spall type classification by hybrid approach” that use models and physical preprocessing for “domain adaptation”. Please extend the literature background.
  3. The classification module uses SVM (if I got it right). The SVM uses the latent vector difference. It seems later in the text that this component is referred to as a “shallow fully connected layer”. Do I get it right? Can you please clarify it in the text?
  4. Gaussian prior using is a standard approach. However, in the current case, to my opinion, more justification is needed. Why it is suitability for vibration data? Do you test other types such as β-VAE or mixture priors?
  5. If I get it right, you are averaging the source and target latent vectors (z and z′). To my opinion it is underexplained. Why do you average them instead of selecting / weighting dynamically?
  6. As far as I understand the process you described in the paper, the test-time training is adapting only the encoder. Do you also tested the impact of adapting the classifier jointly? Or do you test in a teacher-student setting?
  7. Do you average the results were averaged over multiple runs? Do you use seed? If yes, what seed did you used?

Author Response

Comment1 : There is a known problem of test-training leakage in the field of condition based maintenance by vibration analysis because the number of different types of faults in public available dataset is very low (e.g., CRWU contains less than 20 different real fault shapes). Why this paper does not have this problem?

Response1 : Thank you for this important observation. We fully agree that train-test leakage is a critical issue in condition-based maintenance studies, especially when relying on limited or repetitive fault types from single-source datasets like CWRU. To address this, we constructed a benchmark using four diverse public datasets collected under different machines, fault types, sensor setups, and acquisition environments. Furthermore, we clearly separated the source and target domains in our experimental design: the VAT dataset was exclusively used for testing and was never included during training. This structural separation ensures that no fault instances or operating conditions from the target domain are leaked into training. We have added a clarification on this point in Section 4.1.5, Integration Setting and Result.

Comment2 : The paper does not include literature background on other approaches of zero-fault learning such as “Zero-fault-shot learning for bearing spall type classification by hybrid approach” that use models and physical preprocessing for “domain adaptation”. Please extend the literature background.

Response2 : Thank you for pointing out this important issue. We also recognize that, in the field of zero-fault learning, hybrid frameworks that combine model-based approaches with physics-informed signal preprocessing are playing a very important role in domain adaptation. Accordingly, we have expanded the literature review in this paper to include these hybrid approaches, and have added the relevant content to the DA background section. These additions have been reflected in Section 2.3 of the revised manuscript.

Comment3 : The classification module uses SVM (if I got it right). The SVM uses the latent vector difference. It seems later in the text that this component is referred to as a “shallow fully connected layer”. Do I get it right? Can you please clarify it in the text?

Response3 : Thank you for pointing this out. You are correct in your interpretation. While the conceptual framework is inspired by SVM-style margin-based classification in latent space, the actual implementation uses a single fully connected (FC) layer that performs the same linear decision function with SVM. To avoid confusion, we have revised the manuscript to clarify this distinction. Specifically, we added an explanation in Section 4.3.1 to explicitly state that the classifier is implemented as a single FC layer for end-to-end training, while maintaining the intended behavior of an SVM.

Comment4 : Gaussian prior using is a standard approach. However, in the current case, to my opinion, more justification is needed. Why it is suitability for vibration data? Do you test other types such as β-VAE or mixture priors?

Response4 : Thank you for your insightful comment. We agree that the suitability of the Gaussian prior should be more explicitly justified. In our framework, we adopt a standard Gaussian prior for the VAE latent space due to its analytical tractability and stable behavior in unsupervised settings. More importantly, since our model is designed for source-free unsupervised domain adaptation, where no source data or class labels are available during adaptation, more expressive priors such as β-VAE or mixture models would require additional supervision or structural information, which is incompatible with our constraints. For this reason, we opted for a standard normal prior as the most practical and stable choice for learning transferable representations from vibration data. We have added a clarifying explanation in Section 2.2 of the revised manuscript.

Comment5 : If I get it right, you are averaging the source and target latent vectors (z and z′). To my opinion it is underexplained. Why do you average them instead of selecting / weighting dynamically?

Response5 : Thank you for your valuable comment. We agree that further clarification on the rationale for averaging the source and target latent representations is needed. Averaging combines the generality of the source domain with the instance-specific adaptation from the target domain, providing robust representation under distribution shifts. It also acts as a soft regularization, preventing overfitting and catastrophic forgetting during test-time adaptation. We will include this clarification in Section 3.3.2, step 2 of the revised manuscript.

Comment6 : As far as I understand the process you described in the paper, the test-time training is adapting only the encoder. Do you also tested the impact of adapting the classifier jointly? Or do you test in a teacher-student setting?

Response6 : Thank you for your valuable comment. According to prior studies, updating only the encoder during test-time training while jointly learning a self-supervised task has been shown to be effective. Based on these findings, we adopted the same approach in our work. Accordingly, we employed a model architecture that integrates self-supervised learning during test-time training. We also confirm that we have not tested our method under a teacher-student setting, as our primary focus was on building a lightweight and effective encoder-adaptive test-time training framework for industrial time-series fault diagnosis. We will include this clarification in Section 4.3.2 of the revised manuscript.

Comment7 : Do you average the results were averaged over multiple runs? Do you use seed? If yes, what seed did you used?

Response7 : The results presented in the paper were obtained by fixing the random seed to 42 to ensure reproducibility. Therefore, it was not necessary to perform multiple runs, as the experimental results can be consistently reproduced under the same conditions. We will include this clarification in Section 4.3.1 of the revised manuscript.

Round 2

Reviewer 1 Report

Comments and Suggestions for Authors

ACCEPT

Reviewer 2 Report

Comments and Suggestions for Authors

The authors addressed all my comments.